# Design of efficacious somatic cell genome editing strategies for recessive and polygenic diseases

Jared Carlson-Stevermer[1,2,6], Amritava Das[1,3,6], Amr A. Abdeen[1], David Fiflis[1,2], Benjamin I Grindel[1,2], Shivani Saxena[1,2], Tugce Akcan[4], Tausif Alam[4], Heidi Kletzien[2], Lucille Kohlenberg[1], Madelyn Goedland[1,2], Micah J. Dombroe[1] & Krishanu Saha [1,2,5 ✉]

Compound heterozygous recessive or polygenic diseases could be addressed through gene correction of multiple alleles. However, targeting of multiple alleles using genome editors could lead to mixed genotypes and adverse events that amplify during tissue morphogenesis. Here we demonstrate that Cas9-ribonucleoprotein-based genome editors can correct two distinct mutant alleles within a single human cell precisely. Gene-corrected cells in an induced pluripotent stem cell model of Pompe disease expressed the corrected transcript from both corrected alleles, leading to enzymatic cross-correction of diseased cells. Using a quantitative in silico model for the in vivo delivery of genome editors into the developing human infant liver, we identify progenitor targeting, delivery efficiencies, and suppression of imprecise editing outcomes at the on-target site as key design parameters that control the efficacy of various therapeutic strategies. This work establishes that precise gene editing to correct multiple distinct gene variants could be highly efficacious if designed appropriately.

[1] Wisconsin Institute for Discovery, University of Wisconsin-Madison, Madison, WI, USA. [2] Department of Biomedical Engineering, University of Wisconsin-Madison, Madison, WI, USA. [3] Morgridge Institute for Research, Madison, WI, USA. [4] Department of Surgery, University of Wisconsin-Madison, Madison, WI, USA. [5] Retina Research Foundation Kathryn and Latimer Murfee Chair, Madison, WI, USA. [6] These authors contributed equally: Jared Carlson-Stevermer, Amritava Das. ✉email: ksaha@wisc.edu

Gene therapies typically involve the editing of a single allele[1], or delivery of exogenous genetic material (through nucleic acid delivery, viruses, or ex vivo engineered cells) to overexpress the gene of interest or suppress translation of the defective allele[1]. The development of these strategies traditionally starts with studies in animal models, however, such studies are frequently insufficient for genetic disorders that are polygenic—involving different mutations in different alleles that exacerbate a diseased phenotype. Animal models for polygenic diseases are challenging to generate, and the number of mutations implicated in a particular disease precludes the generation of animal models for every mutation (e.g., >400 for *GAA*, the causative gene for Pompe disease[2]). Animal models for genetic disorders have different genomes from that of humans and thus interrogating human gene therapy in animal backgrounds may leave some questions unanswered, especially those regarding off-target effects of editing strategies. Therefore, as delivery systems improve[3,4] and genome editors become more precise[5–17], new platforms and approaches (Fig. 1a) are needed to fully understand the genotypic and phenotypic implications of editing multiple alleles within a person or a human cell.

Genome editors are routinely evaluated against a single allele within a pool of cells. Such studies have produced robust methods to understand the genomic changes, downstream gene expression, and phenotypic changes from editing a single allele, as has been demonstrated in several prior studies[18–23]. Some of these studies, however, reveal that unintended genomic deletions and translocations are potential outcomes at the on-target allele[24], and such unintended outcomes are predicted to be exacerbated as when making multiple cuts in the genome through the delivery of two or more different genome editors[25]. In addition, the adeno-associated viral vectors (AAVs), commonly used to deliver genome editors, can integrate into 5% of targeted alleles[26]. Finally, clonal analysis of genome editing outcomes demonstrates that precise editing of a single mutant allele can generate unintended mutations in other alleles[27]. Thus, the genomic integrity and allelic composition of cells could be variable when attempting to edit multiple alleles, leading to variable expression of multiple alleles with a single cell. To date, precise correction of multiple alleles and an associated phenotypic rescue has not been demonstrated within a single human cell.

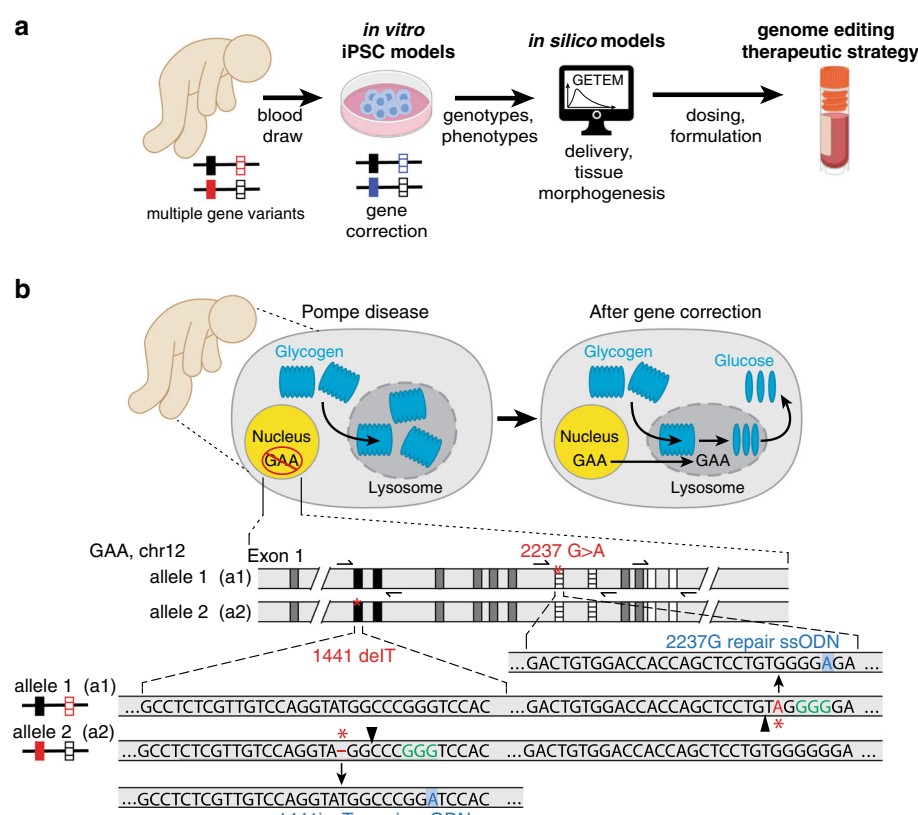

**Fig. 1 A combined in vitro and in silico strategy to evaluate the efficacy of different gene correction therapeutic strategies. a** Schematic indicating the modeling approach in which samples from patients are collected ex vivo are then genome-edited in vitro. Genotypes and phenotype outcomes from the in vitro studies are inputs for an in silico model that simulates the delivery of the therapeutic in vivo as well as tissue morphogenesis (GETEM model, Fig. 5). The results of the in silico model can ultimately guide dosing and formulation decisions. **b** Editing strategy for gene correction of Pompe-diseased induced pluripotent stem cells (iPSCs). Pompe disease is caused by two defective copies of the acid-α-glucosidase (*GAA*) gene. This enzyme is responsible for breakdown of glycogen within lysosomes inside cells. Without GAA, glycogen build up can cause downstream health issues. After correction, *GAA* expresses a functional protein leading to a reduction in glycogen. The schematic indicates the editing locations within *GAA* locus and CRISPR gene correction strategy. In the Pompe patient-derived line, cells harbor compound heterozygous mutations in *GAA*. Allele one, a1, contains a point mutation that causes a premature stop codon (*GAA*:c.[1441=2237G>A]) while allele two, a2, carries a one base pair deletion (*GAA*:c.[1441delT;2237=]). For the CRISPR gene correction strategy, single guide RNAs (sgRNAs; the predicted DNA double-strand break by *Spy*Cas9 is denoted by the arrowhead) were designed to be specific to only the diseased allele by containing the mutant bases (red) within the seed region. Single-stranded oligonucleotides (ssODNs) used for genomic repair contained the wildtype sequence at the mutation site (blue) as well as a silent mutation "wobble" to remove the PAM site (green) to prevent re-cutting of the corrected allele while preserving the amino acid sequence of *GAA*.

Here we determine the phenotypic consequences of correcting single or multiple pathogenic mutations within a single patient-derived cell. We focus on infantile-onset Pompe disease, an autosomal recessive glycogen storage disorder caused by multiple mutations in the acid-α-glucosidase (GAA) gene. GAA encodes an enzyme that breaks down glycogen within the lysosome[28] (Fig. 1b). Over 400 different GAA mutations have been noted within ClinVar, and detailed case studies indicate a buildup of glycogen, leading to clinical complications, most prominently in cardiac and muscle tissues[2]. Left untreated, patients with infantile-onset Pompe disease typically die within the first year of life, and Pompe disease is now frequently included within newborn screening panels[29]. While enzyme replacement therapy (ERT) using recombinant human GAA (rhGAA) and other gene and cell therapies are in development for Pompe disease[30–35], none of these approaches retain endogenous GAA regulation nor have corrected the underlying GAA mutations (Supplementary Information). Once some of the consequences of gene correction of multiple alleles within a single cell are characterized, important questions remain regarding how to design translational studies with gene correction strategies, both for in vivo somatic gene editing strategy or for ex vivo cell therapy with autologous gene-corrected cells.

Patient-derived induced pluripotent stem cells (iPSCs) harbor the exact mutations to be targeted by editing strategies and can recapitulate some aspects of cell and tissue pathology within affected patients[36–39]. Mathematical and computational tools can provide insight on somatic processes at scales larger than cells and small tissue constructs[40], such as genome editor delivery, tissue morphogenesis, and physiological responses (Fig. 1a). Many genetic diseases do not require the correction of all diseased cells: for example, 44% of a live donor pancreas[41] could treat a patient with type 1 diabetes (with event-free survival of both the donor and the recipient[42]), and targeted correction of <1% of tissue progenitors may be sufficient to address various diseases[21,22]. If precise editing of multiple alleles is feasible within a single cell, then a combined in vitro–in silico approach (Fig. 1a) could help answer open translational questions regarding the efficacy of allele targeting and genome editor delivery/dosing strategies.

Using CRISPR–Cas9 genome editors, we first characterize the phenotypic consequence of genome correction of two alleles within the same cell and phenotype rescue of the disease phenotype in both iPSC cells and in a differentiated therapeutically relevant tissue. We then elucidate the second step of a platform for polygenic diseases (Fig. 1a) by generating a quantitative in silico model to determine an optimal therapeutic strategy by considering several mechanisms: differing rates of progenitor and mature cell proliferation, precise and imprecise editing at two alleles, and enzymatic cross-correction mechanisms. We trained and validated our in silico model by considering results from 47 mice across nine studies utilizing four gene therapy delivery strategies and six different genome editors. We then adapted this model to examine cell therapy using ex vivo engineered cells, and the therapeutic phenotype and persistence of the engrafted cells. Our analysis indicates that correcting multiple alleles in vitro is possible and can correct phenotypic defects via both single and double gene correction. For cell therapy, the phenotype of the ex vivo engineered cells and their persistence relative to native tissue determines the efficacy of ex vivo genome-engineered cell therapy. Additionally, in diseases where non-cell-autonomous mechanisms can be therapeutic (e.g., storage disorders with enzymatic cross-correction), precise correction strategies can be highly efficacious. For these conditions, several genome editing strategies[43,44] could be designed with different delivery systems[45,46] and higher fidelity genome editors to correct multiple mutant alleles within a particular patient.

## Results

### Correction of two distinct diseased alleles within Pompe iPSCs.

To correct two endogenous alleles within the same cell, several clonal isogenic iPSC lines were generated by CRISPR–Cas9 gene editing of an iPSC line derived from a patient with infantile-onset Pompe disease[47]. In this line, compound heterozygous GAA mutations responsible for the disease phenotype are a deletion of a thymidine nucleotide at position 1441 (GAA:c.[1441delT], "1441delT") causing a frameshift, and premature stop codon on one allele, and a G>A conversion at nucleotide 2237 (GAA:c.[2237G>A], "2237G>A"), forming an immediate stop codon on the other (Fig. 1b). The mutations within GAA in this patient are ~6.1 kb apart, and hence using a single DNA double-strand break (DSB) with homology directed repair from a long plasmid or viral donor would likely be inefficient[17]. We, therefore, used a strategy utilizing two distinct SpyCas9 ribonucleoproteins (RNPs) with accompanying single-stranded oligonucleotide (ssODN) templates encoding the gene correction (Figs. 1b and 2a and Supplementary Tables 1 and 2). The use of a transient RNP-based strategy lowers the lifetime of the editor within the cells and therefore reduces the chance of off-target and adverse events[48].

Using a combination of S1mplex[19] and ArrayEdit[49] technologies developed by our lab, we enriched for properly-edited iPSCs after delivery of the two genome editors by tracking the presence of genome editors within the nucleus (Fig. 2b). Next, by using high-content analysis imaging of the iPSC clones during culture post-delivery of the editors, we tracked the growth rate of clones, as well as screening the pH of the lysosome[47] using a Lysosensor dye. Lysosensor is sensitive to the buildup of glycogen in the diseased lysosome of mutant GAA cells, as high glycogen phosphorolysis neutralizes this otherwise acidic organelle[47] (Supplementary Fig. 1 and Fig. 2a, b). Individual plotted colonies were also assayed for the presence of either genome editor (Fig. 2c, represented in either purple or green), both genome editors (Fig. 2c, red) and low amounts of genome editors (Fig. 2c, black). Colonies of interest for subsequent analysis were identified as those with high genome editor expression and lower growth rates, presumably arising from the stress of genome editing.

We isolated cell lines that were corrected at the 1441delT allele and the 2237G>A allele individually (termed 'single-corrected'). Single corrected clones remain identical to the unedited line at the unedited locus and contained PAM wobble on the corrected allele (Fig. 2d). Sequencing chromatograms do not show evidence of undesired NHEJ products. Wobble A bases in the corrected lines are highlighted in Fig. 2d to indicate repair from the ssODN. We also isolated a clone corrected at both GAA:c.[1441delT];[2237G>A] alleles (Fig. 2d, termed 'double-corrected'; clone 'c1'). The double-corrected line contained PAM wobble at both loci. (Supplementary Fig. 2 for characterization of the corrected lines. After karyotyping each of the isolated lines, we observed no large transversions or inversions (Fig. 2e) and verified that all gene-corrected lines remained pluripotent (Fig. 2f). Because genome editing can create large indel mutations[50], we also conducted an 8 kb PCR on GAA that included both sgRNA target sites and observed no genomic deletions between the sgRNA target sites (Fig. 2g, h). Sequencing of these large PCR amplicons confirmed that both alleles were present, and no other sequence abnormalities were detected at the edited loci (Supplementary Fig. 2). Finally, chromatograms from Sanger sequencing at the top ten off-target sites for each sgRNA matched the untransfected, patient-derived cell line, indicating that none of the top ten off-target regions were modified by our editing strategy (Supplementary Figs. 3 and 5; Supplementary Table 4). To ensure that our observations were not specific to one clone, five additional double-corrected lines (Supplementary Fig. 7; clones c21, c28,

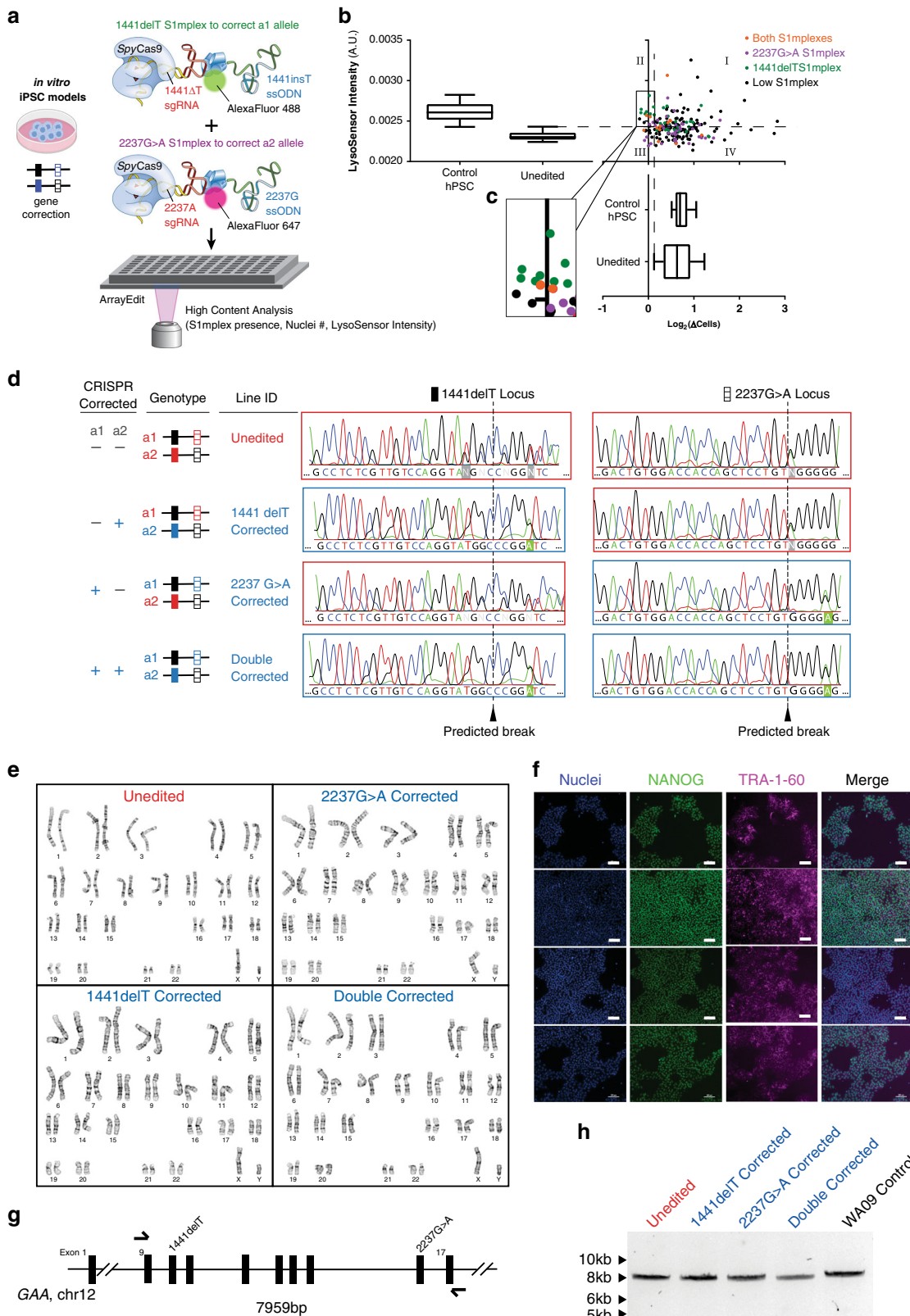

c29, c30, and c73) were generated using the S1mplexes shown in Fig. 2c or by a transient puromycin treatment protocol[27].

Quantitative RT-PCR (qRT-PCR) at the 3' end of the *GAA* mRNA transcript, as well as around each edited locus (Fig. 3a, Supplementary Fig. 6), indicated that the corrected loci were correctly expressed. We observed that the unedited line expressed the lowest levels of *GAA* transcripts when compared to internal *GAPDH* levels (Supplementary Fig. 6), despite the presence of full-length, mature mRNA that could be used to express the protein. Both the single- and double-corrected lines also expressed mature *GAA* transcripts. By looking for the presence of disease variants and protoadjacent motif (PAM) wobbles

**Fig. 2 Gene correction of two distinct mutant alleles within a single human cell. a** ArrayEdit-based isolation of iPSC clones corrected at either or both loci. CRISPR S1mplex design for the gene correction of compound heterozygous mutations. S1mplexes targeting 1441delT mutant were labeled with an AlexaFluor488 compound while S1mplexes targeting the 2237G>A mutation were labeled with an AlexaFluor647. These ribonucleoprotein genome editors were mixed prior to transfecting into cells and subsequently plated on the ArrayEdit platform to conduct high-content analysis (see Supplementary Fig. 1). **b** Left: LysoSensor quantification per μFeature of two mock transfections after 7 days of growth. Normal control hPSCs were significantly more intense than unedited, Pompe diseased iPSCs on ArrayEdit. Bottom right: the growth rate of unedited and control hPSCs following a mock transfection to establish a baseline for growth. Growth rates were calculated by measuring the per-day change in the number of cells of the μFeature. Top right: LysoSensor intensity was plotted against growth rate per μFeature to identify edited colonies. Dashed lines indicate regions of interest. ($n = 145$ independent cell lines). **c** Magnification of quadrant II from (**b**). μFeatures in this region were selected for genomic analysis to isolate edited clones. ($n = 17$ independent cell lines). **d** Sanger sequencing traces of corrected cell lines. The unedited line contains mutations at both alleles: 1441delT mutation causes a breakdown of sequence trace, whereas a single point mutation demonstrates a heterozygosity 2237G>A locus. *Spy*Cas9 cut site is denoted by a dotted line. **e** Karyotypes of all isolated gene-corrected lines as well as unedited cells. No abnormalities were detected at a band resolution of 500. **f** Immunocytochemistry of pluripotency markers in gene-corrected lines. All lines were positive for pluripotency markers NANOG and TRA-1-60 (scale bar: 100 μm). **g** Schematic of long PCR covering both *Spy*Cas9 cut sites. Arrows denote primers. The expected PCR amplicon is 7959 bp in length. **h** Gel analysis of long-range PCR described in (**g**) in each isolated cell line. No significant deviances from the expected length were detected, and no other notable bands were observed. WA09 control cells are hPSCs.

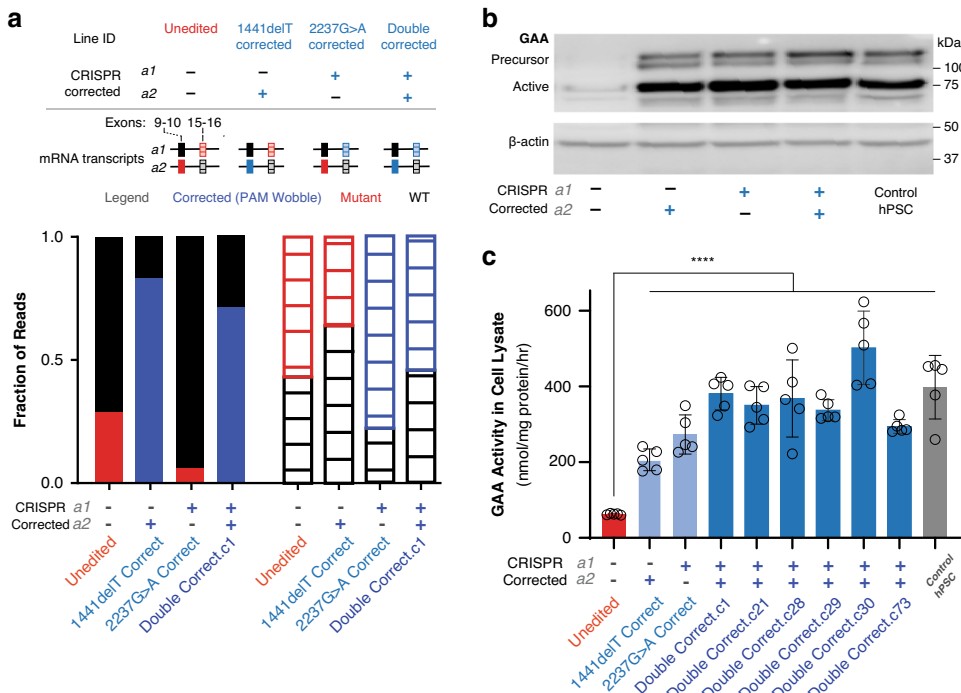

**Fig. 3 Transcription and protein levels within gene-corrected cells. a** Schematic indicating the genotypes of the iPSC lines generated and mRNA transcripts collected. Deep sequencing from qRT-PCR around both diseased loci in isolated cell lines. Reads were classified as WT, mutant, or corrected (PAM wobble) and mapped to either allele 1 (top bar) or allele 2 (bottom bar). When neither allele was corrected, both alleles were expressed at approximately the same rate. However, when either mutation was corrected, the corresponding allele was expressed at a higher rate than the one that still possessed a mutation. When both alleles were corrected, the fraction of reads making up the population was evenly distributed. Observations at individual alleles were consistent across both assayed loci. **b** Western blot for GAA protein. Each of the corrected lines expressed high levels of active protein as well as detectable levels of the precursor protein. Unedited cells expressed significantly lower levels of GAA protein, but the level was still above the limit of detection. **c** GAA activity in cell lysate as measured by 4-MUG cleavage in acidic conditions. Unedited cells have significantly lower activity showing there was little to no active protein. Corrected cell lines include Double Correct.c1 (isolated via ArrayEdit), Double Correct.c21, Double Correct. c28, Double Correct.c29, Double Correct.c30 (isolated via sequential correction of 2237G>A allele followed by S1mplex electroporation correction of 1440delT) and Double Correct.c73 (isolated via sequential correction of 2237G>A allele followed by using transient puromycin based correction of 1440delT). All corrected lines had significantly higher activity than unedited cells but were indistinguishable from each other ($n = 5$ technical replicates, ****$p = 3.59 \times 10^{-12}$, two-tailed $t$-test, $\alpha = 0.05$, heteroscedastic; mean ± s.d.).

introduced by the ssODN (Fig. 1b) via deep sequencing on endpoint PCR samples of mRNA, we observe that both alleles are expressed individually at higher levels (3–5 fold increase) than unedited cells (Supplementary Fig. 6). Each allele is expressed similarly to the corresponding single corrected line (Supplementary Fig. 6b, Supplementary Information). These findings suggest nonsense-mediated decay of the mutant

transcript[51] or cellular compensation[52] to overcome the mutant allele within the single corrected lines. We detected active GAA protein using a Western blot (Fig. 3b) at levels comparable to a control hPSC line. We were also able to identify precursor polypeptides, which are important for protein secretion[53], showing the *GAA* transcripts from the edited alleles are correctly translated and processed within cells. Notably, we were able to

detect only small amounts of GAA protein and precursor polypeptides in the unedited iPSCs. All edited cell lines (Supplementary Fig. 7) were able to produce (Fig. 3c) and secrete active GAA (Fig. 3d and Supplementary Fig. 6d).

**Enzymatic cross-correction by gene-corrected cells**. Detection of active GAA secretion by the edited cells led us to test the potential of edited cells to enzymatically cross-correct diseased cells (Fig. 4a). Because Pompe disease has a significant effect on cardiac tissue in infants, we differentiated iPSC lines to cardiomyocytes (Pompe iPSC-CMs) using a previously described small-molecule inhibitor protocol[54] (Fig. 4a). For all differentiations, we observed spontaneous contraction (Supplementary Videos) and confirmed the expression of α-actinin, a marker of cardiac lineage commitment (Supplementary Fig. 8a). Similar to results seen in the iPSC state, differentiated corrected lines still expressed and secreted active GAA, as indicated in a 4-MUG cleavage assay on cardiomyocyte protein lysates and spent culture media (Supplementary Fig. 8b–e, Supplementary Information). It has previously

been demonstrated that by culturing in medium devoid of glucose, Pompe iPSC-CMs display an accumulation of glycogen within the lysosome[47]. We performed a medium exchange experiment wherein we took a partially spent, glucose-free medium from each corrected line (putatively containing secreted active GAA) and used it to replace glucose-free medium on unedited Pompe iPSC-CMs (Fig. 4a). One day after this media exchange, cells were stained with LysoSensor, and subsequent confocal microscopy was used to measure lysosome acidity as a proxy for glycogen clearance. As a control, we added rhGAA to unedited Pompe iPSC-CMs to simulate ERT. When unedited cardiomyocytes were supplemented with 10 nM rhGAA (ERT), LysoSensor intensity increased, indicating clearance of glycogen from the lysosome. Media from all edited cells were able to recover the lysosomal pH at 96 h (Fig. 4b), and this clearance is expected to continue until normal levels of glycogen were reached[55]. Within these cultures, we qualitatively observed lysosomal size of unedited Pompe iPSC-CMs in GAA-positive media through visualization of Lysosomal Associated Membrane Protein 1, (LAMP-1). Detailed quantification of lysosome size was not

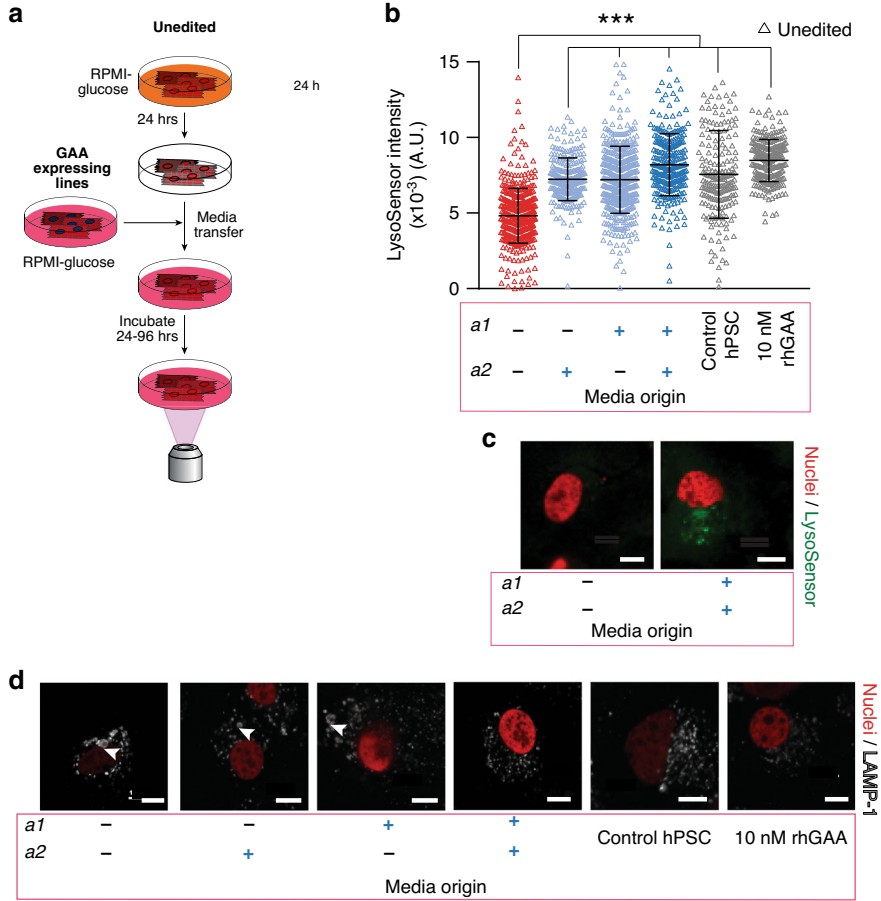

**Fig. 4 Enzymatic cross-correction of unedited cardiomyocytes by gene-corrected cardiomyocytes. a** Schematic of enzymatic cross-correction experiments using gene-corrected cardiomyocytes. Unedited iPSC-CMs (red) were supplied media without glucose for 24 h (orange). After 24 h, media was replaced with media (pink) that had previously been exposed to corrected cell lines (blue) or supplemented with rhGAA. 24–96 hours after replacement, unedited cells were stained with LysoSensor and imaged using confocal microscopy for dye intensity. **b** Quantification of LysoSensor intensity in cross-corrected lines 96 h post media exchange. Each triangle represents a corrected cell identified using CellProfiler. After 96 h of daily media changes or supplementing with rhGAA, all conditions had a significant increase in dye intensity over control conditions. Unedited cells were modified to express histone 2B (H2B)-mCherry to facilitate imaging of the nuclei in these assays. See also Supplementary Fig. 7. (***$p < 10^{-15}$, $n = 262, 201, 261, 249, 135$, and 302 independent lysozymes respectively per condition, as detailed in Supplementary Table 7, two-tailed $t$-test, $\alpha = 0.05$, heteroscedastic; mean ± s.d). **c** Representative images of unedited iPSC-CMs stained with LysoSensor in media from unedited and double-corrected iPSC-CMs. **d** Representative images of LAMP1 staining in unedited, single corrected, double-corrected cells and control PSC-CM and unedited iPSC-CM treated with rhGAA. (scale bars: 10 μm). Select enlarged LAMP1-positive lysosomes are identified by white arrowheads.

possible, as the pH-sensitive dye utilized in this method is prone to photobleaching as noted previously[56]. In media from unedited cells, lysosomes were enlarged, consistent with the buildup of glycogen (Fig. 4d). In comparison, when media was taken from double-corrected cells or supplemented with rhGAA, lysosomes appeared as punctae. Samples from single corrected cells fell between these two extremes. Taken together, the single- and double-corrected cells enzymatically cross-correct diseased cardiomyocytes quickly and effectively.

**Establishing a Pompe disease gene therapy efficacy model.** Having observed phenotypic rescue with single- and double-correction in both iPSCs and disease-relevant, differentiated cardiomyocytes, we next sought to identify potential strategies to design translational studies for polygenic diseases. For instance, determining the minimum effective dose of genome editors for infants with Pompe disease involves the consideration of tissue morphogenesis, delivery to various tissues, edited protein levels, as well as the spectrum of various genomic outcomes. Whenever possible, we look to patient data to gain information about these processes and accordingly have created a in silico approach to model somatic cell genome editing. Specifically, to investigate the dynamics of gene correction approaches in vivo, we developed an in silico GEne Therapy Efficacy Model (GETEM) simulating genome editing of two distinct alleles within a developing liver. The gene-corrected liver within Pompe diseased patients would act as a depot for GAA dissemination to distal organs (e.g., the heart and skeletal muscle[30]), based on mechanisms described in prior gene augmentation studies in animal models and currently in clinical trials[57].

First, we simulated standard ERT to ground our model with empirical clinical data gathered from infants and children treated with ERT[58]. This grounding enables our Pompe GETEM to account for enzymatic cross-correction (Fig. 5a) properly. Despite high levels of GAA in the liver on ERT, GAA transport to and uptake within the distal muscle can be low[59], and this is accounted for in Pompe GETEM through a loss factor (Supplementary Information). Liver progenitors in this model are proliferative and give rise to non-proliferative mature cells[60]. Both progenitor and mature cells can absorb extracellular GAA to be enzymatically cross-corrected from a diseased to a normal state, and subsequently reverted to the diseased state as GAA degrades. In distal cardiac and skeletal muscle tissues, the tissue absorbs GAA for enzymatic cross-correction to a normal phenotype (Supplementary Information). Biweekly intravenous ERT doses[58] lead to an oscillatory percentage of phenotypically normal cells in the heart, skeletal muscle, and liver within our model (Fig. 5b, Supplementary Fig. 9), reaching 32.5% normal cells in the heart on average after one year, similar to levels observed with ERT, enough for significant heart glycogen clearance[59].

**Empirical genome editing data to ground and validate GETEM.** Because genome editing data of the human *GAA* within patients in vivo or within any animal model are not available, we use empirical data from previously published genome editing experiments targeting a single endogenous gene with mice. These studies provide quantitative preclinical data that ground the delivery mechanisms to the liver within GETEM. While several studies have performed precise gene correction studies in the mouse livers, they have typically delivered their editor of choice using a plasmid[46,61], a virus[62] or an encapsulated mRNA (in conjunction with a virus for the donor correction template)[63]. In GETEM, we consider all these delivery strategies to produce RNPs that ultimately produce precise and imprecise genome edits

within a cell (Fig. 5c). We then utilized the published empirical data from plasmid DNA, mRNA and viral delivery to train GETEM ($n = 18$ mice for training, three different genome editors), and subsequently validated GETEM using data from RNP delivery into the liver[64] and four more liver editing studies[65–68] ($n = 29$ mice for validation; Fig. 5d; details on mouse studies in Supplementary Table 8).

To account for the different delivery strategies, a previously reported model of the central dogma[69] was employed to calculate the amounts of transcribed and translated RNP. Either (1) plasmid DNA transcription and mRNA translation, (2) mRNA translation, or (3) AAV-mediated transcription and translation led to the formation of editor proteins. In comparison to sgRNA transcription, Cas9 transcription or translation was assumed to be rate limiting for the formation of the RNP when Cas9 is delivered as an mRNA or encoded on DNA. AAV doses were converted to an effective viral DNA dose based on the published vector copies per cell data provided[62]. The degradation of the plasmid or viral DNA occurs constitutively as previously measured with mouse macrophages[69]. For studies that targeted the *Fah* allele, a selection pressure enriches for edited cells Fig. 5e. The dynamics for the enrichment has been well characterized through prior experimental studies[70]. For these studies, the unedited cells die quickly unless the mouse is continuously fed 2-(2-nitro-4-trifluoro-methylbenzoyl)-1,3-cyclohexanedione (NTBC). Once corrected, the liver cells can survive without NTBC supplementation. Within our model, delivery of genome editors to livers within $Fah^{-/-}$ mice can target a mutant fumarylacetoacetate hydroxylase allele to correct a mutated stop codon. Both $Fah^-$ progenitor and mature hepatocytes can be edited to form $Fah^+$ hepatocytes, and imprecisely edited $Fah^-$ hepatocytes. The $Fah^-$ hepatocytes are subject to a death rate which is modified by the presence of NTBC, and thus this model incorporates the growth advantage of precisely edited cells.

GETEM considers the rate of genome editing, $\frac{dE}{dt}$, to be a second-order mass action rate equation dependent on both the number of RNPs, $N_{RNP}$, and the number of cells, $N_{Cell}$ (Eq. (1)), where k is the genome editing rate constant:

$$\frac{dE}{dt} = k.N_{Cell}.N_{RNP} \qquad (1)$$

The change of edited cells against time, $\frac{\Delta E}{\Delta t}$, was calculated using numerical differentiation for each time step (Eq. (2)):

$$\frac{dE}{dt} \sim \frac{\Delta E}{\Delta t} = \frac{E_n - E_{n-1}}{t_n - t_{n-1}} \qquad (2)$$

To account for tissue growth, this time derivative was normalized for the total number of cells in the mouse at the same time point and averaged over the experiment duration (Eq. (3)):

$$\overline{\frac{dE}{dt}} = \overline{\frac{1}{N_{Cell}} \frac{\Delta E}{\Delta t}} \qquad (3)$$

For each published study in the training dataset ($n = 18$ mice), random sampling of the experimental data by bootstrapping[71] produced ten thousand samples that constituted a surrogate "bootstrapped" data set. For each of these bootstrapped samples, the average rate of liver genome change, $\overline{\frac{dE}{dt}}$ was calculated individually (Fig. 5d). The mean and standard deviation of this set of bootstrapped $\overline{\frac{dE}{dt}}$ values was utilized in GETEM to simulate the range of outcomes for the mice in each study in the training set (Fig. 5e, left four panels).

Training studies employed three different genome editors delivered using four different methods: plasmid delivery of *Spy*Cas9[61], gRNA, and repair template in a $Fah^{-/-}$ mouse model;

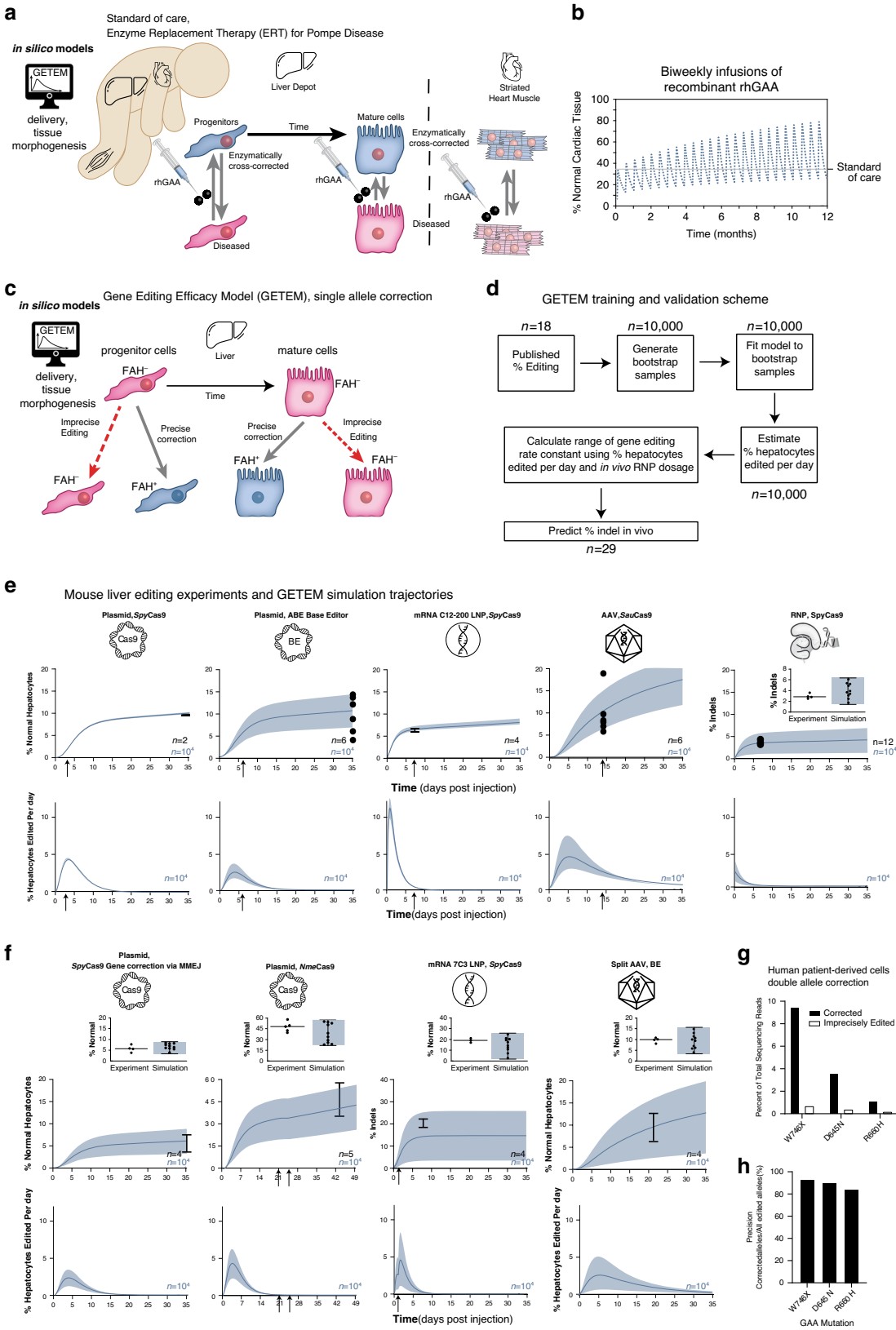

**a** Standard of care, Enzyme Replacement Therapy (ERT) for Pompe Disease

**b** Biweekly infusions of recombinant rhGAA

**c** Gene Editing Efficacy Model (GETEM), single allele correction

**d** GETEM training and validation scheme

**e** Mouse liver editing experiments and GETEM simulation trajectories

**f**

**g** Human patient-derived cells double allele correction

**h**

plasmid delivery of base editor and gRNA in a $Fah^{-/-}$ mouse model[46]; mRNA delivery of $Spy$Cas9 (C12-200 lipid nanoparticle) and gRNA and AAV delivery of repair template in a $Fah^{-/-}$ mouse model[63]; and, AAV delivery of $Spy$Cas9, gRNA, and repair template to edit Ornithine transcarboxylase ($Otc$)[62]. Ten thousand simulated samples were generated from the empirical

data for each mouse via bootstrapping[71], and the GETEM was fit to each of these bootstrapped samples to calculate mean ± 1 s.d. for the gene-editing rate constant via Eqs. (1)–(3). The trained GETEM shows strong concordance of the simulation with experimental data at the assay timepoints for each study (black experimental data is within the blue range of simulation

**Fig. 5 Derivation, empirical training, and validation of in silico gene therapy efficacy model (GETEM) for Pompe disease correction in a developing infant. a** Schematic showing enzymatic cross-correction of somatic tissues by injected rhGAA (enzyme replacement therapy, ERT). For a given developing liver of a patient, progenitor cells proliferate and can asymmetrically divide into mature cells. Injected rhGAA is absorbed by the liver as well as striated muscle tissue (both heart and skeletal muscle). **b** Percentage of normal cardiac tissue within a heart of a Pompe diseased infant after one year of biweekly ERT at 20 mg/kg. Percentages of normal skeletal muscle tissue and hepatocytes are in Supplementary Fig. 9. **c** Schematic for in silico gene editing model for previously published precise correction of a single base pair in mouse disease models, **d** Flowchart for training and validating the GETEM model using previously published studies of in vivo somatic cell genome editing of a mouse liver by intravenous injection. **e** (top) Absolute change of edited cells over time predicted by the GETEM for the studies utilized in (**d**). For studies that employed selection in the liver, arrows indicate the time at which the selective pressure favoring edited cells was applied (for $Fah^{-/-}$ models, this indicates the removal of NTBC supplementation in the diet; and, for *Otc* editing, this indicates the induction of a high protein diet.). **f** Validation of the model using additional published editing strategies. Arrows on days 21 and 24 indicate removal of selection pressure on the treated mice as per published experimental protocol. Arrow on day 2 in the mRNA LNP indicates redosing of the LNP. (*n* represents biological replicates as previously reported in the literature, mean ± s.d). **g** Percentage of total sequencing reads from primary fibroblasts of Pompe diseased patients that were treated with S1mplexes targeting W746X D645N or R660H GAA mutations. Results indicate gene correction and imprecise editing for three different mutations. **h** Percentage of edited alleles that are precisely edited in (**g**).

outcomes). The lines in Fig. 5e indicate the mean and standard deviation of edited alleles in the mice cohorts, while dots signify the percentage edited alleles of individual mice for studies in which individual replicate values were reported.

To validate the model, the bootstrapped $\frac{dE}{dt}$ values from all the training studies were combined. The mean and one standard deviation of the combined set of surrogate $\overline{\frac{dE}{dt}}$ values were used to simulate the genome editing outcomes for RNP doses employed by a separate study[64] outside of the training dataset (Fig. 5e, right). There is strong concordance of the experimental data with the results of the simulation at the assay timepoint, day seven (Fig. 5e, inset), as the percent of edited alleles at the assay timepoint (black) nearly all fall within the range of predicted percent of edited alleles (blue). Lower panels in Fig. 5e, f indicate the rate of liver genome editing for each individual bootstrapped sample ($n = 10{,}000$ per panel) over time for the various mRNA, plasmid, and AAV studies.

To further validate GETEM, we modeled other somatic cell genome editing strategies evaluated in mouse livers—microhomology based gene correction using *Spy*Cas9[65], *Neisseria Meningitidis* Cas9 (*Nme*Cas9) based editing[66], a different type of lipid nanoparticle (7C3 mRNA Lipid Nanoparticle) delivering *Spy*Cas9 mRNA[67] and AAV delivery of split base editors BE3[68]. The simulation results for all these strategies showed concordance with the experimental data (Fig. 5f). In total, the validation data set includes 29 mice across four different delivery strategies with four different editing strategies (Fig. 5d–f).

The increase in the percentage of edited cells is dependent on both genome editor activity and genome editor independent processes (growth dynamics, selection of edited phenotypes). However, in vivo, these processes can be difficult to analyze separately. Using the validated GETEM approach, we can analyze the processes of liver genome change and genome editor activity separately. The top panels (Fig. 5e, f) indicate the cumulative *change* in the liver genome over time. The lower panels (Fig. 5e, f) exhibit the *rate* of the genome editing in the target cells, hepatocytes. During the duration of the simulation (top panels, Fig. 5e, f), the percentage of the liver genome that is changed continues to rise even after genome editor activity declines (bottom panels, Fig. 5e, f). Genome editor activity increases initially for all genome editors except RNP. This is driven by transcription and translation of the delivered payload to generate genome editors within cells in situ. For dosing with RNP genome editors, there is no in situ generation of the editor within the liver, and only a decline in activity is observed. Thus, the prolonged increase of editing at the tissue-level highlights the role of genome editor independent processes (e.g., attributed to growth or selection of edited phenotypes) in driving efficacy.

In order to extend GETEM—already validated for single allele editing—to biallelic gene correction for Pompe disease, additional training data is required on the precision of biallelic genome editing and their cellular phenotypic consequences. We performed additional in vitro experiments with four different pathological mutations in *GAA* in Pompe patient-derived cells (Fig. 5g, h). Precise gene correction with S1mplexes occurred approximately in 84–93% of the edited sequence reads at the target locus as assayed by deep sequencing of genomic DNA. The remaining are imprecise, unintended modification to the on-target site, which could destroy the PAM or modify the on-target site for subsequent editing of these alleles. According to the nomenclature of Shen et al.[72] to describe this ratio of gene correction to other editing outcomes, S1mplexes are approximately "precise-80." In comparison, normal *Spy*Cas9 is precise-50, and prime editors[44] are precise-90. Next, experimental measurements of functional outcomes from this distribution of edited cells ground protein levels in the Pompe GETEM. Cells corrected at both alleles a1 and a2 have been modeled to secrete the same amount of GAA than those edited at a single allele (Fig. 3d) at first approximation. We include a parameter in our cell therapy GETEM to explore the effects of higher secretion arising from double-corrected cells. In summary, high-resolution measurements of alleles after genome editing, as well as quantitative measures of protein levels and activity post-editing, provide key empirical rate constants for the training of Pompe GETEM.

**Design of in vivo treatment informed by Pompe GETEM.** The power of building a computational model is that many different doses can be quickly simulated, providing insights into the spectrum of genotypes with the tissues over time as well as with the effects of this gene-corrected tissue on other parts of the body. Building on the model in Fig. 5, we generated a model that considers ERT along with gene editing outcomes after the administration of genome editors targeting multiple alleles (Fig. 6a). First, we simulate doses of genome editors required to achieve or exceed efficacy equivalent to ERT standard of care. For S1mplexes, the dosages of RNP to reach comparable efficacy as ERT were evaluated using the editing rate established in Fig. 5. The S1mplex dose was evaluated to be 23.9 mg/kg/allele to reach equivalent healing in the liver using 6 monthly doses via intrahepatic injections, starting at birth, scaled using the 50th percentile growth of a male infant. This dose represents the amount of S1mplex that reaches the hepatocytes, and not the amount that may need to be injected systemically. At one year after the first injection, the normal cells in the heart reach 32.5%, the standard calculated from the ERT simulation (Fig. 6b), and the normal cells in the liver consist of 34%

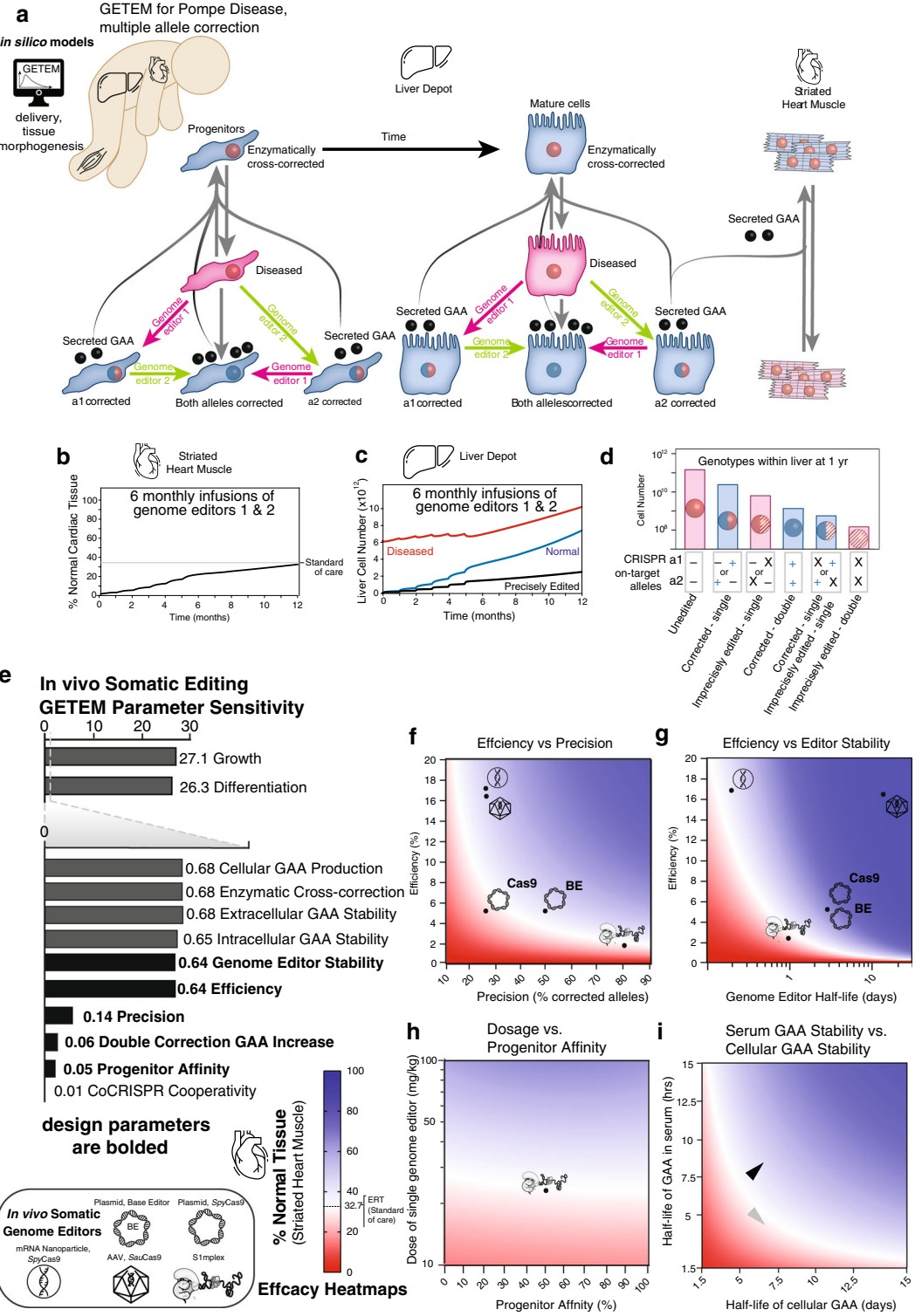

genotypically corrected cells (corrected at alleles a1, a2 or both, Fig. 6c), and 66% enzymatically cross-corrected cells (Fig. 6c). Genome editing post addition of genome editor is a relatively quick process, and the percentage of edited cells stabilizes within five days (Supplementary Fig. 10). However, enzymatic cross-correction via secreted and absorbed GAA is a slower process, and continues beyond the five day gene-editing period (Supplementary Fig. 10). Single-corrected cells constitute the most prevalent edited genotype (Fig. 6d).

Sensitivity analysis (Fig. 6e) indicated that tissue morphogenesis factors and cell/tissue biology factors also control efficacy (Supplementary Information). These intrinsic biological properties of each patient may offer opportunities to tailor therapies to each infant. The sensitivity analysis helps clarify trade-offs required for gene therapy decisions. For instance, when considering increasing the magnitude of benefit by editing both alleles, or editing one allele with higher efficiency—a lower sensitivity is observed for increasing GAA production within a

**Fig. 6 In vivo somatic cell gene correction strategies involve tradeoffs between efficiency, precision, progenitor affinity, and editor stability. a** In silico Gene Therapy Efficacy Model (GETEM) for Pompe disease correction in a developing infant. Schematic showing gene correction for two diseased alleles in a liver indicating correction of alleles, a1 and a2, by genome editors 1 and 2 to form gene-corrected cells capable of secreting GAA to enzymatically correct other unedited cells. Secreted GAA is also absorbed by striated muscle tissue (both heart and skeletal muscle). **b** Percentage of normal cardiac tissue within a developing heart of a Pompe diseased infant after the administration of six doses of genome editors at 23.9 mg/kg. **c** Cell numbers indicating growth of diseased, normal, and precisely-edited cells in the gene-edited liver depot after the administration of 6 doses of genome editors at 23.9 mg/kg. **d** Distribution of genotypes in the gene-edited liver depot after the administration of 6 doses of genome editors at 23.9 mg/kg. **e** Sensitivity analysis of the model indicating the absolute values of parameter sensitivity of tissue morphogenesis factors, genome editor factors, and cell/tissue biology intrinsic factors. **f** Tradeoff between genome editor efficiency and genome editor stability, focusing on the percentage of enzymatically cross-corrected heart tissue. Heatmap indicates that lower efficiency editors could be efficacious if the extracellular editor stability increases. **g** Tradeoff between genome editor efficiency and precision, focusing on the percentage of enzymatically cross-corrected heart tissue. Heatmap indicates that lower efficiency editors can be efficacious if higher precision editors are used. **h** Tradeoff between increasing genome editor dose and progenitor affinity, focusing on the percentage of enzymatically cross-corrected heart tissue. **i** Using GETEM, heatmap indicating tradeoff in heart muscle correction in the developing infant between the degradation rates of GAA in the serum and cellular GAA, indicating that stabilization of GAA in the serum can improve clinical outcome (gray arrowhead indicates pre-stabilization, black arrowhead indicates post-stabilization). Simulation results for liver and skeletal muscle are shown in Supplementary Figs. 11 and 12.

double-corrected cell (0.06) relative to a single-corrected cells than editing efficiency or precision (~0.64). This lower sensitivity indicates that editing a single allele at higher efficiency is likely to be more efficacious, as a sensitivity of 0.06 implies that a 1% increase in GAA production caused by biallele editing would lead to a 0.06% increase in percentage of normal cardiac tissue, whereas, the higher sensitivity of editing efficiency and precision of 0.64 implies that a 1% increase in either editing efficiency or precision would increase the percentage of normal cardiac tissue by 0.64%. Therefore, for S1mplexes, there is higher expected benefit in editing a single allele more efficiently, than in editing both alleles at a lower efficiency. In addition to growth and differentiation rates of the edited cells, outcomes are highly sensitive to the efficiency, stability, precision, and progenitor affinity of the genome editor. These later parameters relevant to the design of the somatic cell genome editors are bolded (Fig. 6e).

Systematic in silico investigation of thousands of genome editing strategies are summarized in several heatmaps (Fig. 6f–i and Supplementary Figs. 11 and 12). First, we observe that imprecise editors require much higher efficiencies to achieve similar therapeutic levels in the heart (32.7% normal; Fig. 6f and Supplementary Figs. 11 and 12). Using a precise-80 editor similar to the S1mplex, even though it has an efficiency of 1%, can produce similar outcomes as *Spy*Cas9, which is almost six times more efficient, but only precise-25. Editing strategies used for model validation in Fig. 5e are also displayed on the heatmap, and these include *Spy*Cas9 delivered as a plasmid[61], *Spy*Cas9 delivered as an encapsulated mRNA with an AAV HDR template[63], *Sau*Cas9 delivered via AAV[62] and RA6.3 base editor delivered as a plasmid[46]. Second, we observe a negative correlation between editor efficiency and genome editor half-life required to reach therapeutic levels in the heart (Fig. 6g and Supplementary Figs. 11 and 12). The parameter for efficiency in our model lumps diverse processes of cellular uptake of the editors, intracellular trafficking, nuclease activity and DNA repair at the target site because few empirical studies have been able to distinguish these processes in vivo. The efficiency of S1mplex editors at this dose level was calculated to be 0.29–2% based on the bootstrap sampling based method presented in Fig. 5. Decay rate refers to the extracellular degradation or binding by serum proteins or other cells/biomolecules in the extracellular space that prevent the activity of a genome editor. Therefore, strategies that stabilize the editor in the extracellular space—even with low (~1%) efficiency editors[73]—could be one strategy to increase efficacy. Third, lower dosing with delivery strategies targeting progenitors could be a potent feasible strategy with highly-precise, yet low-efficiency strategies (e.g., base editors[45] and S1mplexes[74]): increasing the

progenitor affinity reduces the combined genome editor dose by 50% for comparable efficacy (Fig. 6h and Supplementary Figs. 11 and 12). Progenitor targeting can, therefore, increase the potency of a genome editing-based gene therapy approach. Fourth, interventions that affect progenitor growth rate, enzymatic cross-correction, and serum GAA stability (e.g., adjuvants like Duvoglustat[75]) could promote efficacy. Increasing serum GAA stability by altering serum GAA half-life from 4 h (Fig. 6i and Supplementary Figs. 11 and 12, gray arrowhead) to 8 h (Fig. 6i and Supplementary Figs. 11 and 12, black arrowhead) could increase the percentage of corrected cardiac tissue by 50%. Therefore, gene therapy with GAA stabilizing adjuvants may also provide a way to boost efficacy with low-efficiency editors. Finally, we recognized that there is variation within clones in their ability to secrete GAA (Fig. 3c), and therefore modeled the therapeutic levels in the liver (Supplementary Fig. 12b), assuming that double-corrected liver cells produce between 0 and 150% excess GAA than single corrected cells. In these simulations, there is a minimal difference due to the excess secretion of GAA, due to the small number of double-corrected cells (Fig. 6d).

Because single-edited genotypes outnumber double-edited genotypes in our GETEM results, dosing time and dosing amount are not critical in the design of therapy, as we observe limited differences in potency between a single large dose or multiple smaller doses (Supplementary Fig. 13, Supplementary Information). Since genome editors can trigger an immune response[76–79], which likely scale with the dose[80,81], we have performed our simulations using multiple smaller doses of genome editors rather than a large single dose. The sensitivity analysis allowed us to consider tradeoffs in genome editor selection, progenitor targeting, genome editor delivery, and GAA stabilization via an adjuvant.

**Design of gene-corrected cell therapy with GETEM.** Autologous cell therapies involving gene-corrected cells avoid exposure of the body to genome editors in vivo and, therefore, may be a preferred strategy over in vivo editing, if an immune response to the editor and off-tissue or off-target effects in the body present serious safety issues. Editing both loci could provide additional therapeutic benefit, but the therapeutic benefit could depend on whether cells engraft well and retain similar function after transplantation into the body. GETEM simulations for autologous cell therapy was implemented for various cell therapeutic strategies: single-corrected progenitor cells (proliferative), double-corrected progenitor cells (proliferative), as well as single-corrected mature cells (non-proliferative) and double-corrected

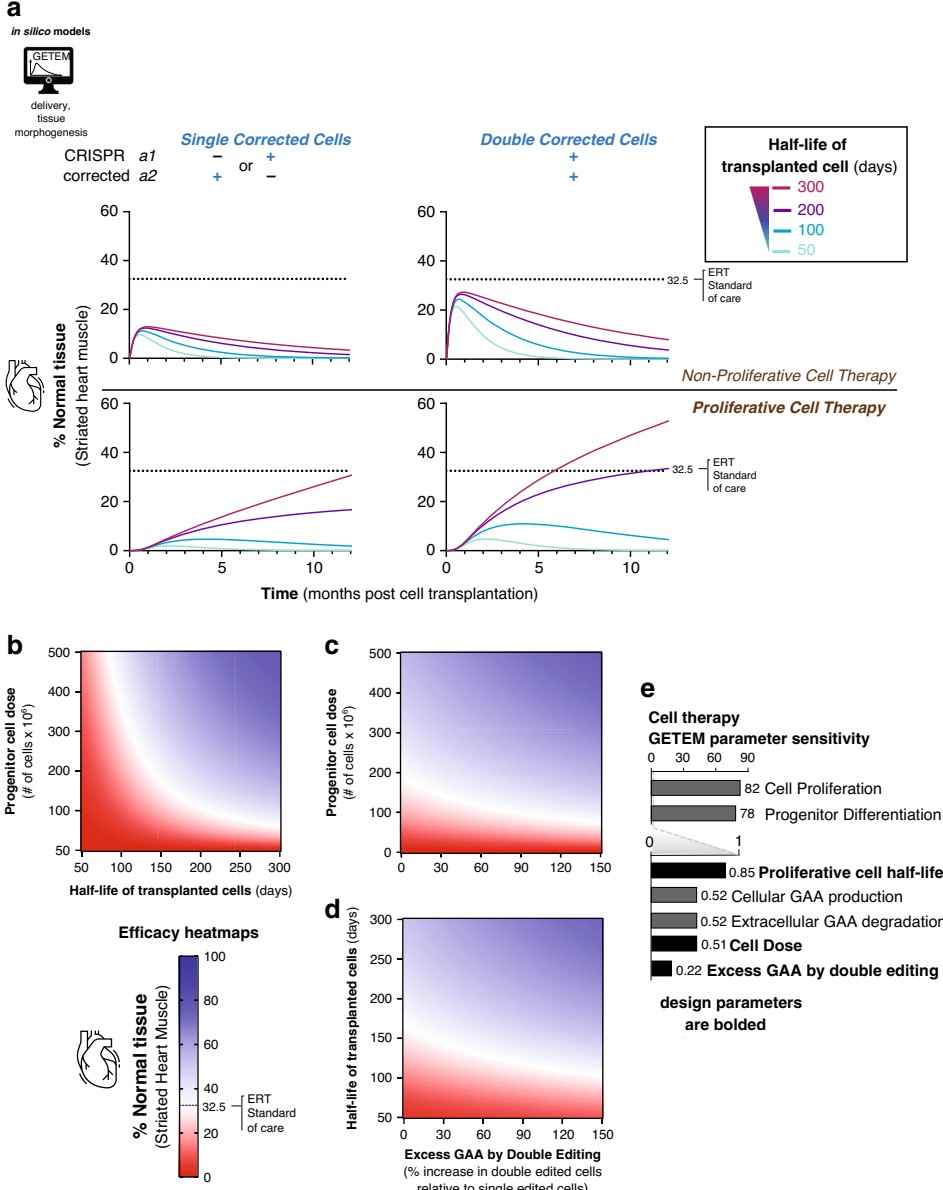

**Fig. 7 Durable gene-corrected cell therapy for Pompe disease requires persistent proliferative cells. a** Plots showing the degree of phenotypically normal cardiac tissue when either 10 billion non-proliferative single- or double-corrected cells are dosed or 0.1 billion proliferative single- or double-corrected cells are dosed. For these plots, the GAA production for double-corrected cells is 150% excess that of single corrected cells for four different half-lives of dosed cells. For long-lasting correction, proliferative progenitor cells need to have a half-life exceeding 100 days. **b** Heatmap showing the proliferative cell dose (assuming double-corrected cells with 75% excess GAA production compared to single-corrected cells) against the half-life of dosed cells. If the dosed cells have a half-life similar to endogenous hepatocytes, a 75 million cell dose is sufficient for matching ERT. **c** Heatmap showing the proliferative cell dose (assuming that the dosed cells have 250-day half-life) against the GAA production rate of dosed cells, demonstrating that dose has a higher effect on therapeutic efficacy than GAA production gained from double correction. **d** Heatmap showing that the GAA production against the half-life of the dosed cells (assuming that 250 million cells were dosed), indicating that the half-life of the dosed cells has a higher effect on therapeutic efficacy than the GAA production gained from double correction. **e** Sensitivity analysis for model parameters relevant to cell therapy. In addition to growth and differentiation rates of the transplanted cells, outcomes are highly sensitive to the proliferative cell half-life, cell dose and excess GAA produced by double gene correction. These later parameters, relevant to the design of therapy with autologous gene-corrected cells, are bolded.

mature cells (non-proliferative). The half-life of engrafted functional cells was systematically varied from 50 days to 300 days, the typical half-life of a hepatocyte[82]. For cell therapy with mature cells, no doses up to $10 \times 10^9$ cells could achieve efficacy comparable to ERT (Fig. 7a). However, cell therapies with proliferative progenitor cells could result in durable, efficacious responses for a single $0.1 \times 10^9$ cell dose if the half-life of an engrafted progenitor cell exceeds 100 days. While there is

detectable therapy for cells that engraft and live for shorter periods of time, there is not a durable response, and redosing would likely be necessary. Double-corrected cells provide an added benefit for a similar dose of cells (Fig. 7a, assuming that double-corrected cells secrete 150% excess GAA than single corrected cells) versus the same dose of single corrected cells.

To understand dosing strategies for cell therapy, we performed thousands of simulations varying the proliferative progenitor cell

dose, half-life, and GAA production (Fig. 7b–d). Sensitivity analysis on cell therapy GETEM indicates that tissue morphogenesis factors were much more sensitive (Fig. 7e) in comparison to cell therapy factors (dose, lifespan, and secretory phenotype of engrafted cells), or cell/tissue intrinsic factors (extracellular degradation or single allele GAA production rate). Even if the dose is raised to $500 \times 10^6$ proliferative progenitors, if the lifespan is less than 100 days, a therapeutic outcome matching ERT is not achieved (Fig. 7b). Similarly, even if the dosed cells have native hepatocyte lifespans, as few as $75 \times 10^6$ cells are required to observe a therapeutic effect matching ERT (Fig. 7b). If double-corrected cells engrafted well and have 75% higher production of GAA compared to single corrected cells, the dose required to reach the ERT therapeutic threshold is halved (Fig. 7c, assuming a 250-day half-life of the transplanted cells). Finally, engrafting cells with higher GAA secretion will match the standard of care, even if the lifespan of cells is lower, establishing a tradeoff between high GAA secretion and persistence of transplanted cells (Fig. 7d). Similar effects were observed on the striated skeletal muscle tissue (Supplementary Fig. 14) and the liver depot (Supplementary Fig. 15). These results combining our experimental data with GETEM support the conclusion that ex vivo cell therapy could be efficacious in addressing pathogenic polygenic mutations. Therefore, while in vivo dosing of genome editors is expected to deliver therapeutic effect largely through the involvement of single-corrected cells (Fig. 6d), a double correction strategy would be preferable for ex vivo cell therapy as it would reduce the dose of cells required. Finally, precisely edited cells in these models could also represent cells transduced in vivo with a gene augmentation therapy. Overall, we believe GETEM could be easily adapted for a wide variety of somatic tissues treated with various cell and gene therapies.

## Discussion

Many polygenic diseases have no animal models available for preclinical studies[18], because many animal models arise from a single gene disruption. Moreover, it is currently infeasible to generate unique transgenic animal models for every patient's specific mutation, and it is difficult, if not impossible, to recapitulate the human genetic background in animal models. In contrast, human iPSCs provide an important biological substrate to evaluate genome editing approaches, as data from iPSC studies have been used to inform human clinical trials for gene therapies[83,84]. Using allele-specific genome editors, we demonstrate biallelic gene correction of two distinct mutations in a single iPSC. Many of the common Pompe disease mutations can be targeted in an allele-specific manner using *Spy*Cas9 strategies (Supplementary Table 6). We observe that transcriptional regulation is driven by the endogenous promoter, potentially correcting many different isoforms for *GAA*[85,86]. The tissue targets of genome editors, therefore, could expand from the traditional foci of liver and muscle to other tissues that may use alternate *GAA* isoforms. In contrast, in standard gene augmentation approaches, all cells must process a single isoform. Further, silencing from synthetic or viral elements has been observed for gene therapies, and in our hands with targeted knock-in strategies that overexpress a transgene via a synthetic promoter (Supplementary Fig. 8f). Transgene silencing raises concerns about the durability of viral gene therapies and proposed cell therapies where *GAA* is overexpressed from a safe harbor locus[35]. In our gene correction strategy, post-translational processing of the enzyme also appears to be intact, as the distribution of processed GAA is identical to healthy controls. In contrast, *GAA* over-expression in mammalian cells can cause cellular stress, leading to differential trafficking and processing of the nascent translated

peptide[53]. Furthermore, an mRNA- or RNP-based gene correction strategy reduces insertional oncogenesis by using a non-viral approach for the delivery of the genome editor.

Our analysis indicates that the precision of genome editing is a key intrinsic parameter controlling efficacy for in vivo somatic editing approaches. Guide RNA design for CRISPR genome editors has already been identified as critical in the design of genome editors, and the genome editing community now routinely employs design tools integrating empirical and computational approaches[72,87–89]. Our approach complements these guide RNA design tools to consider mechanisms in vivo involving delivery, cell targeting, and stability of editors as well as cell biological processing of the edited protein product. Mechanistic modeling of the genotypic outcomes of gene correction within a growing, differentiating tissue is necessary to understand the somatic impact of genome editing. For animal studies and clinical studies, GETEM provides insight into the distribution of edited genotypes within cells in the body and could be tailored to an individual animal or patient, creating "digital twins" of specific bodies undergoing somatic cell genome editing. Our in vitro and in silico results come together to inform cell therapeutic approaches, in which patient cells are ex vivo engineered to produce GAA from the native locus, as opposed to a safe harbor locus or viral vector[90]. Both single and double gene correction are efficacious for in vivo somatic cell editing strategies. While the predicted benefits from correcting both alleles are small within simulations of in vivo editing, these benefits are greater for the design of cell therapies. GETEM establishes a quantitative basis for making tradeoffs between GAA secretion, persistence, and dose of engrafted cells while considering therapeutic effects in three different organs. Thus, further development of GETEM could facilitate rapid in silico evaluation of different strategies for both gene therapies and cell therapies at an organismal level. Such an expanded toolkit may ultimately reduce the number of preclinical studies needed to establish efficacy before embarking on first-in-human clinical trials for genome editing therapeutics.

For gene correction in the liver, we evaluated a variety of strategies using plasmid delivery using hydrodynamic injection (an approach not easily translatable to larger animals[91]) viral methods (delivering both the editing machinery or just the correction DNA template), which carry significant risks of viral integration[26], mRNA delivery and ribonucleoprotein delivery. Our simulations were able to design efficacious dosing and formulation for Pompe disease patients for all of these strategies. When analyzing base editor editing outcomes in vitro[46] (with no selection for edited outcomes, RA6.3 base editor) at the on-target site, 45% of the on-target adenines were converted to guanines, but, 55% of the adenines upstream were also converted to guanines. Therefore, the on-target precision is about 50%, or precise-50 using the Shen et al.[72] nomenclature, as all the other adenine to guanine edits are on-target, but imprecise. Prime editing[44] addresses this issue, and future development with in vivo somatic delivery of base and prime editors has strong potential even at low delivery efficiencies.

The GETEM framework is versatile and can simulate a variety animal studies involving a wide range of growth characteristics and selection pressures for a wide array of editing strategies. For example, in their demonstration of *Nme*Cas9 to introduce indels to knock out *Hpd* within a mouse Hereditary Tyrosinemia Type 1 model[66], the authors observe that *Hpd* indel cells are metabolically reconditioned such that they survive dietary tyrosine despite the knock-out of *Fah*. Additionally, the mice used in this experiment lost 20% of their body weight in 15 days and were between 15 and 20 weeks of age. The growth and differentiation rate constants in our GETEM were easily modified to reflect slow growth in these mice. The authors also reported that they fed

mice with NTBC from days 21 through 24 of the experiment, which can be modeled in our GETEM, and it has effects on the rate at which the fraction of the edited hepatocytes in the liver grows (Fig. 5f). Finally, the plasmid DNA dose was calculated using the length of *Nme*Cas9 + sgRNA plasmid (4790 bp), which is significantly shorter than the modified pX330 plasmid used in other *Spy*Cas9 studies (8484 bp). The breadth of studies successfully simulated by our GETEM approach indicates that it can likely be modified easily to accommodate selection, enzymatic cross-correction, and biallelic editing for many different future gene therapy applications.

Multiple characteristics of a particular somatic cell genome editing therapeutic strategy are analyzed in GETEM: the enzymatic editing rate of the editor itself, the editor's precision, the delivery of the editor, and the selection of edited and unedited outcomes. When reporting the results of in vitro and in vivo experiments, these results are coupled and are reported as a single value indicating either the percentage of alleles edited, or the percentage of cells demonstrating phenotypic correction. Our GETEM approach decouples these constituent characteristics, enabling individual modeling of these characteristics or selectively grouped modeling of these characteristics to discover synergistic effects within these processes. It enables us to analyze how each of these interact with each other and affect the overall therapeutic outcome and suggest trade-offs for gene therapy improvement. The monotonous growth of the percentage of edited cells in cases with and without selection of the edited phenotype (Fig. 5e, f, top plots), contrasts with the dramatic rise and fall of the percentage of the liver undergoing editing (Fig. 5e, f, bottom plots). This contrast highlights the editor-agnostic processes largely driving the growth of the edited phenotype in animal models.

Current limitations of our approach arise from incomplete knowledge of several phenomena, including immune response, adverse effects linked to off-target modifications, transport of editors and edited protein products, and tissue morphogenesis. First, an immune response to the editor, delivery vector, or the edited protein product is possible. Continuous constitutive expression of *Sau*Cas9 in hepatocytes elicits an innate immune response, which accelerates the death of the edited hepatocytes[92]. Using strategies with a shorter-term expression of genome editor (mRNA) may prevent adverse events in the edited cells. For Pompe disease, patients already exhibit a variable response to the ERT, leading to variable efficacy. Both transient and long-term immunosuppressive strategies can be employed in conjunction with in vivo somatic editing and cell therapeutic approaches to mitigate these immune responses, and future work to expand GETEM could leverage empirical studies on the immune response both at the cellular level in vitro or in vivo. Second, we did not isolate any iPSC lines that harbored detectable off-target events in the top ten predicted sites, but others in the field have noted cells to undergo phenotypic changes induced by cutting the genome, such as cell cycle arrest[93] and upregulation of innate immune transcriptional programs[94,95]. Further, more complex events like large deletions and translocations could occur at the on- and off-target site that would be missed by our Sanger sequencing approach[50]. The chromatin state of iPSCs may also alter the ability of off-target sites from being edited. However, all of the methods used to characterize off-target modifications are inherently incomplete, as it is not feasible to non-destructively achieve whole-genome sequencing of every single edited cell[96]. Therefore, additional studies in applying the genome editor to the target tissue of interest could be necessary to identify and model additional off-target events. Further characterization of more clones and within differentiated cells[95] could shed light on these low-frequency events, which become crucial for cell therapeutic approaches where upwards of $10^8$ cells have now been implanted

into patients[97]. Third, correction efficiencies may also only be lower in post-mitotic cells with some genome editors, and dispersion within tissues may be inhomogeneous, leading to spatial patterns of edited cells/progeny that could have variable phenotypic effects. Finally, additional empirical studies of the pharmacokinetics of editors and protein products, liver growth rates during development, glycogenolysis, and engraftment of edited cells could help us refine several assumptions made in GETEM (see Supplemental Information). Overall, these limitations provide insights into the types of measurements that are needed—detailed genomic analysis, cellular analysis, transport measurements and morphogenesis measurements—to enable a more predictive platform.

Overall, our results indicate that with appropriate engineering and design, recent advances in base editing[46], delivery vectors[98], nuclease decoration[99], and nuclease engineering offer additional possibilities to establish efficacious gene correction therapies in a streamlined fashion with reduced dependence on animal models. We, therefore, anticipate that multiple dosing with safe and precise genome editors can be developed for a greatly expanded set of targets in diseases with compound heterozygous and complex polygenic origins.

## Methods

**Cell culture**. All human pluripotent stem cells (hPSCs) were maintained in mTeSR1 medium on Matrigel (WiCell) coated tissue culture polystyrene plates (BD Falcon). Cells were passaged every 4–5 days at a ratio of 1:8 using Versene solution (Life Technologies). Patient-derived iPSC line, Pompe GM04192, was a gift from the T. Kamp and M. Suzuki (UW-Madison). Cardiomyocytes derived from hPSC and iPSC cultures were maintained in RPMI/B27 on Matrigel (WiCell) coated polystyrene plates (BD Falcon). Patient-derived fibroblast lines were obtained from Coriell Institute with different GAA mutations (W746X mutation was from Coriell ID: GM04912; D645N mutation was from Coriell ID: GM20090; R660H was from Coriell ID: GM13522) and cultured in DMEM supplemented with 10% FBS and 1% Penicillin/Streptomycin. All cells were maintained at 37 °C in 5% $CO_2$, and tested monthly for possible mycoplasma contamination. Samples obtained from the Coriell Institute have been approved by the Working Group of the Human Genetic Cell Repository.

**Cardiomyocyte differentiation**. hPSCs and iPSCs were differentiated into cardiomyocytes using a small molecule-directed differentiation protocol in a 12-well plate format[54]. Briefly, all adherent hPSCs and iPSCs were dissociated in TrypLE solution (Life Technologies), counted with a hemocytometer, and centrifuged at $200 \times g$ for 5 min. Cells were plated at a density between 0.5–1 × $10^6$ cells/well depending on cell line. Once tissue culture plate wells reached 100% confluency (day 0), medium in each well was replaced with a solution containing ml RPMI/B27-Insulin (Life Technologies), 12 μM CHIR99021 (BioGems 25917), and 1 μg/ml Insulin solution (Sigma-Aldrich I9278). Exactly 24 h later (day 1) medium in each well was removed and replaced with RPMI/B27-insulin. Exactly 48 h after (day 3) half of the spent medium was collected. To this, an equal volume of fresh RPMI/B27-Insulin was mixed. This combined media was then supplemented with 7.5 μM IWP2 (BioGems 75844). Two days later (day 5) medium in each well was replaced with RPMI/B27-Insulin. Two days (day 7) later and every three days following, spent medium was replaced with RPMI/B27. Spontaneous contraction was generally observed between days 12–16 of differentiation.

**Creation of S1m-sgRNAs to correct both mutant alleles**. S1m-sgRNAs were designed by combining the sgRNA scaffold with the S1m aptamer sequence[74]. S1m gBlocks were annealed with Phusion polymerase (New England Biolabs) under the following thermocycler conditions: 98 °C for 30 s followed by 30 cycles at 98 °C for 10 s, and 72 °C for 15 s with a final extension at 72 °C for 10 min. S1m cDNA was annealed with Phusion polymerase (New England Biolabs) under the following thermocycler conditions: 98 °C for 30 s followed by 30 cycles at 98 °C for 10 s, 60 °C for 10 s, and 72 °C for 15 s with a final extension at 72 °C for 10 min. In vitro transcription was performed with the MEGAShortscript T7 Kit (Thermo Fisher Scientific) according to the manufacturer's instructions. For guide RNAs for fibroblast transfection, in vitro transcription was performed using HiScribe T7 RNA synthesis Kit (New England Biolabs) according to the manufacturer's instructions.

**Genome editor delivery**. All hPSC transfections were performed using the 4D-Nucleofector System (Lonza) as per the manufacturer's instructions. Briefly, hPSCs were harvested using TrypLE (Life Technologies) and counted. 2 × $10^5$ cells per transfection were then centrifuged at $100 \times g$ for 3 min. Excess media was aspirated

and cells were resuspended using 20 μl of RNP solution per condition and then electroporated using protocol CA-137. 50 pmol Cas9, 60 pmol sgRNA, 50 pmol streptavidin, and 60 pmol ssODN were used to form particles per ssODN-S1mplex as described previously[74]. Cells were then harvested using TrypLE (Life Technologies) and counted. $2 \times 10^5$ cells per transfection were then centrifuged at $100 \times g$ for 3 min. Excess media was aspirated and cells were resuspended using 20 μl of RNP solution per condition. After nucleofection, samples were incubated in nucleocuvettes at room temperature for 15 min prior to plating into $2 \times 10^4$ cells per well on ArrayEdit in mTeSR media + 10 μM ROCK inhibitor. Media was changed 24 h post transfection and replaced with mTeSR1 medium. Fibroblast transfections were performed in 24 well plates using 50,000 cells/well using 2 μl Lipofectamine 2000/well (0.5 μg Cas9/well and sgRNA, streptavidin and ssODN at a 1:1:1:1 molar ratio).

**Synthesis of ArrayEdit platform.** Microcontact printing was performed to modify the wells of a standard tissue culture plate[100]. The surface modification involved printing of an alkanethiol initiator to nucleate the polymerization of hydrophilic poly(ethylene glycol) (PEG) chains. Briefly, double sided-adhesive was attached to the bottom of a standard tissue culture plate, after which a laser cutter was used to cut out the well bottoms. Patterns were transferred to gold-coated glass via a polydimethylsiloxane stamp after which the glass was submerged in a PEG solution overnight to build PEG chains surrounding μFeatures[101]. Standard tissue culture plates with well bottoms cut out were then fastened to processed sheets using a custom-made alignment device.

**High-content analysis.** Automated microscopy was performed using a Nikon Eclipse Ti epifluorescent scope. A $15 \times 15$ grid with one μFeature per image was established and maintained so that each feature imaged was consistent each day. Nikon Perfect Focus was used to ensure that all colonies were in the same Z-plane and LysoSensor intensity was measured accurately. Images were processed using CellProfiler[102] to count the number of nuclei and quantify LysoSensor intensity.

**DNA Sequencing.** DNA was isolated from cells using QuickExtract DNA Extraction Solution (Epicentre) following TrypLE treatment and centrifugation. Extracted DNA was incubated at 65 °C for 15 min, 68 °C for 15 min, and 98 °C for 10 min. Genomic PCR was performed using AccuPrime HiFi Taq (Life Technologies) or Q5 polymerase (New England Biolabs) and 500 ng of genomic DNA according to the manufacturer's instructions. Long (8 kb) PCR reactions were thermocycled using an extension step of 10 min. Genomic PCR products were then submitted to the University of Wisconsin-Madison Biotechnology Center for DNA sequencing or analyzed on an Illumina Miniseq instrument. Complete list of primers used are available in Supplementary Table 10.

**Off-target analysis.** Candidate off-target sites were identified and ranked for both of the sgRNAs used for gene correction using a previously established algorithm[103]. This algorithm was validated via unbiased genome-wide off-target analysis by Tsai et al.[104], and the top scoring sites are listed in Supplementary Table 4. PCR on genomic DNA from established clonal lines was performed with primers listed in Supplementary Table 3 flanking each of the top ten scoring candidate off-target sites. These PCR products were analyzed using Sanger sequencing by the University of Wisconsin-Madison Biotechnology Center and Genewiz.

**RT- and qPCR.** RNA was isolated from cells using QuickExtract RNA Extraction Solution (Epicentre) following the manufacturer's protocol. 100 ng of extracted RNA was reverse transcribed using Superscript IV Reverse Transcriptase (Invitrogen). Endpoint PCR amplification of cDNA product was performed using 1 μl of cDNA product and AccuPrime HiFi Taq (Life Technologies) following the manufacturer's instructions. Efficacy of the endpoint PCR was evaluated via gel electrophoresis of PCR product in a 1% agarose gel.

qPCR reaction was performed in triplicates for each cell line and sequence (GAPDH, 1441delT, 2237G>A, and GAA), by mixing 10 μl iTaq Universal SYBR Green Supermix (Bio-Rad), 0.5 μl sequence specific forward primer, 0.5 μl sequence specific reverse primer, 1 μl cDNA product, and 8 μl water. qPCR analysis was performed in a CFX96 Real Time PCR System under the following thermocycling conditions: 95 °C for 30 s followed by 35 cycles of 95 °C for 5 s, and 60 °C for 30 s.

**Next generation sequencing analysis.** Cas-Analyzer[105] was used to perform sequence analysis. For each sample, sequences with frequency of less than 1000 were filtered from the data. Sequences in which the reads matched with primer and reverse complement subsequences classified as "target sequences". Target sequences were aligned with corresponding wildtype sequence using global pairwise sequence alignment and analyzed manually for mutants and PAM wobbles.

**Western blotting.** Protein levels of GAA and β-Actin were determined in each cell line. Following cell lysis in ice-cold RIPA buffer supplemented with protease and phosphatase inhibitors and EDTA (5 mM), protein concentration was determined (DC Protein Assay, BioRad). 40 μg of protein from each cell line was loaded into a 4–12% Bis-Tris precast gel (Criterion XT, BioRad) and gel electrophoresis was performed. Proteins were then transferred to a nitrocellulose membrane and blocked in filtered 5% nonfat dry milk in TBS-T (Tris-buffered saline, 0.15% Tween20) for 1 h at room temperature. The membrane was then incubated overnight at 4 °C with GAA (Abcam ab137068, 1:1000) and β-Actin (Millipore, MAB1501, 1:40,000) primary antibodies. Following the incubation period, the membrane was washed in TBS-T and incubated with appropriate horseradish peroxidase secondary antibodies (Goat Anti-Rabbit IgG, Abcam ab205718, 1:2000; Anti-Mouse IgG, Cell Signaling Technologies 7076 1:20,000) for 1 h. The membrane was washed again in TBS-T, and then developed (SuperSignal West Pico Plus Chemiluminescent Substrate, Thermo Scientific) for 5 min using a ChemiDoc-It2 Imaging System (UVP) and imaged.

**GAA Activity assay.** Acid glucosidase activity was measured by hydrolysis of 4-methylumbelliferyl-D-glucoside (4-MUG, Sigma M-9766) at pH 4 to release the fluorophore 4-methylumbelliferone (4-MU)[47]. Briefly, 4-MUG was incubated with 10 μl protein lysate in 0.2 M sodium acetate buffer (pH 4.3) for 90 min at 37 °C. Fluorescence from 4-MU was then measured using a Glomax plate reader (Promega) and activity was calculated using a standard curve.

**Immunocytochemistry.** Live cell imaging of lysosome intensity was done using LysoSensor Green (Life Technologies L7535). Dye was mixed in culture media at a 1:1000 dilution prior to adding media to wells. Cells were then incubated for 5 min in LysoSensor solution. Media was then aspirated and cells were washed 2× with PBS. All imaging was done within one hour of staining.

To assay for pluripotency markers, hPSC cultures were fixed using 4% PFA and incubated at room temperature for 10 min. Cells were then permeabilized using 0.05% Triton X-100 and incubated for 10 min. Following two washes with 5% goat serum, NANOG antibody (R&D Systems AF1997, 1:200) and TRA-1-60 antibody (Millipore MAB5360, 1:150), was added to cells and incubated overnight at 4 °C. The next day, cells were rinsed twice with 5% goat serum and then incubated with a donkey anti-goat secondary antibody (Life Technologies A11055 1:500) for 1 h at room temperature. Cells were then washed twice with PBS and mounted for imaging.

Cardiomyocyte cultures were processed in the same manner as above. After permeabilization cells were incubated with anti-sarcomeric alpha-actinin (Abcam ab68167 1:250) overnight at 4 °C. The next day, cells were rinsed twice with 5% goat serum and then incubated with a goat anti-rabbit secondary antibody (Santa Cruz Biotech sc-362262, 1:500)

**Media exchange.** Cardiomyocytes were cultured in RPMI/B27 + insulin and media was exchanged every 2 days. As a normal media exchange, diseased and corrected cells were introduced to RPMI/B27 +insulin/−glucose. 24 h post change, cells were stained with LysoSensor as described above to determine a baseline fluorescent intensity. After staining, media was replaced with media from either corrected or healthy lines and cultured for an additional 24 h. After incubation, cells were again stained with LysoSensor and imaged using confocal microscopy.

**In silico modeling: gene therapy efficacy model (GETEM).** Preliminary model design was performed using COPASI 4.21[106], and the final model construction was performed using MATLAB R2020a using the SimBiology© package. Ten thousand replicates[107] from published data were generated via the MATLAB (R2020a) bootstrap function using default options in the MATLAB software. The SimBiology sbproj files used for the simulations (Supplementary Table 9) has been provided in the Supplementary Files hosted at Zenodo—https://tinyurl.com/GETEMZenodo (contains Supplementary Videos, Modeling Code, Sequencing Data, and Western blot image).

**Statistics and reproducibility.** For Fig. 2f, three technical replicates performed. For Fig. 2h, PCR was performed once. For Fig. 3b, western blotting was performed once with three technical replicates. For Fig. 4c, d, three technical replicates performed. For Supplementary Fig. 1c, d, three technical replicates performed. For Supplementary Fig. 8, three technical replicates were performed for all conditions.

**Reporting summary.** Further information on research design is available in the Nature Research Reporting Summary linked to this article.

## Data availability

These authors declare that all essential data supporting the conclusion of the study as well as detailed assay protocols, analytical algorithms, and customized computational codes are within the paper and Supplementary materials. Data are present in Zenodo repository (https://doi.org/10.5281/zenodo.3910777, https://tinyurl.com/GETEMZenodo). Code is available at https://github.com/ada586/PompeGetem.git. Raw reads and traces from sequencing are available at NCBI Bioproject PRJNA675893 (https://www.ncbi.nlm.nih.gov/bioproject/PRJNA675893). Any additional relevant information can be obtained from the authors upon reasonable request. Source data are provided with this paper.

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

## Acknowledgements

We thank members of the Saha lab for helpful discussion and comments on the manuscript, Masatoshi Suzuki and Tim Kamp for sharing Pompe iPSC lines, the University of Wisconsin Biotechnology Center for the use of the equipment and technical support, ArtforScience for assistance with figure art, and Aldevron for technical support with Cas9 proteins. We acknowledge generous financial support from the National Science Foundation (CBET-1350178, CBET-1645123), National Institute for Health (1R35GM119644-01), Environmental Protection Agency (EPA-G2013 –STAR-L1), University of Wisconsin Carbone Cancer Center Support Grant P30 CA014520, Wisconsin Alumni Research Foundation, and the Wisconsin Institute for Discovery.

## Author contributions

These authors contributed equally: J.C.-S. and A.D. J.C.-S. and A.D planned research and analyzed data. J.C.-S., A.D., and K.S. designed experiments. J.C.-S., A.D., D.F., A.A., B.G. S.S. and H.K. performed experiments. J.C.-S. and A.D. developed the in silico modeling framework. T. Alam, and T. Akcan provided key insights into design of dosing and delivery for GETEM simulations. J.C.-S., A.D. and K.S. wrote the manuscript with input from all authors. K.S. supervised research.

## Competing interests

J.C.-S., A.A., L.K., and K.S. have filed a patent application on the S1mplex technology. Patent Applicant: Institution - WISCONSIN ALUMNI RESEARCH FOUNDATION (MADISON, WI, US) Name of Inventors: Carlson-stevermer, Jared Matthew (Madison, WI, US) Saha, Krishanu (Middleton, WI, US) Abdeen, Amr Ashraf (Madison, WI, US) Kohlenberg, Lucille Katherine (Madison, WI, US) Application number: United States Patent Application 20180362971 Status of Application: Filed Aspect of Manuscript: Generation of single edited clones, and double-edited clones c1, c28, c29, and c30. The authors declare no other competing interests.
