## [Peer Review File · Nature Communications]

Reviewers' Comments:

Reviewer #1:

Remarks to the Author:

The authors have improved their data quality, and performed additional analysis to address the reviews. All my questions have been adequately addressed.

Reviewer #2:

Remarks to the Author:

CRISPR-based Genome editing provides a novel approach for genetic engineering. In this manuscript, the group used Pompe disease as a model system and presented evidence that genome editors can be designed to target/correct two mutant alleles simultaneously. In addition, the authors carried out in silico modeling for genome editing in vivo, through which they have identified several potentially important parameters for CRISPR genome editing design that can enhance the efficiency of the treatment in the future.

Overall, the work presented in this manuscript is well-planned, carefully executed, and data meticulously analyzed. The conclusions drawn are sensible. However, we have some significant concerns on the manuscript. First, it remains unclear why targeting both alleles is critical for treatment of Pompe disease, which is a recessive disorder, and targeting both alleles in vivo remains technically challenging. The data in this manuscript also show that one allele correction is already sufficient to restore the enzymatic activity to a level close to WT cells. Second, the in silico modeling approach is very interesting, however, there is no experimental support how useful it is in an in vivo model for pompe disease.

Reviewer #3:

Remarks to the Author:

In this revised manuscript, the authors validated the proposed model simulating published genome editing experiments from 47 mice, 6 genome editors, and 4 delivery strategies. In addition, they clarified the key points regarding the editing of one allele or both alleles and the depth analysis of off-target editing events.

The authors have provided adequate responses to my suggestions and concerns.

1 **Response to the Reviewers' Comments– NCOMMS-20-40268-T**

2 Jared Carlson-Stevermer, Amritava Das, et al. "Design of efficacious somatic cell genome
3 editing strategies for recessive and polygenic diseases."

4

5 *Reviewer #1 (Remarks to the Author):*

6 *The authors have improved their data quality, and performed additional analysis to address*
7 *the reviews. All my questions have been adequately addressed.*

8 We thank the reviewer for their positive comments and their time and consideration.

9

10 *Reviewer #2 (Remarks to the Author):*

11 *CRISPR-based Genome editing provides a novel approach for genetic engineering. In this*
12 *manuscript, the group used Pompe disease as a model system and presented evidence that*
13 *genome editors can be designed to target/correct two mutant alleles simultaneously. In*
14 *addition, the authors carried out in silico modeling for genome editing in vivo, through which*
15 *they have identified several potentially important parameters for CRISPR genome editing*
16 *design that can enhance the efficiency of the treatment in the future.*

17

18 *Overall, the work presented in this manuscript is well-planned, carefully executed, and data*
19 *meticulously analyzed. The conclusions drawn are sensible. However, we have some*
20 *significant concerns on the manuscript. First, it remains unclear why targeting both alleles is*
21 *critical for treatment of Pompe disease, which is a recessive disorder, and targeting both*
22 *alleles in vivo remains technically challenging. The data in this manuscript also show that*
23 *one allele correction is already sufficient to restore the enzymatic activity to a level close to*
24 *WT cells. Second, the in silico modeling approach is very interesting, however, there is no*
25 *experimental support how useful it is in an in vivo model for pompe disease.*

26 We thank the reviewer for their consideration of our manuscript. We highlight that single
27 allele correction is sufficient for addressing disease phenotype. We do not claim that editing

28 both alleles is critical for treatment of Pompe disease, and state so in the Discussion “**Both**
29 **single and double gene correction are efficacious for *in vivo* somatic cell editing**
30 **strategies**”. The absence of a published model of human biallelic Pompe disease precludes
31 our ability to test some of the findings *in vivo*, but we believe that the modeling of other *in*
32 *vivo* studies validates our approach.

33

34

35 *Reviewer #3 (Remarks to the Author):*

36 *In this revised manuscript, the authors validated the proposed model simulating published*
37 *genome editing experiments from 47 mice, 6 genome editors, and 4 delivery strategies. In*
38 *addition, they clarified the key points regarding the editing of one allele or both alleles and*
39 *the depth analysis of off-target editing events.*

40

41 *The authors have provided adequate responses to my suggestions and concerns.*

42 We are pleased that our clarifications assisted the reviewer in understanding our paper. We
43 thank them for their time and effort.